# Immune modulatory effect of a novel 4,5-dihydroxy-3,3′,4′-trimethoxybibenzyl from *Dendrobium lindleyi*

Pichayatri Khoonrit[1], Alp Mirdogan[2], Adeline Dehlinger[2], Wanwimon Mekboonsonglarp[3], Kittisak Likhitwitayawuid[1], Josef Priller[2], Chotima Böttcher[2☯*], Boonchoo Sritularak[1,4☯*]

**1** Department of Pharmacognosy and Pharmaceutical Botany, Faculty of Pharmaceutical Sciences, Chulalongkorn University, Bangkok, Thailand, **2** Department of Neuropsychiatry and Laboratory of Molecular Psychiatry, Charité –Universitätsmedizin Berlin, Berlin, Germany, **3** Scientific and Technological Research Equipment Centre, Chulalongkorn University, Bangkok, Thailand, **4** Natural Products for Ageing and Chronic Diseases Research Unit, Faculty of Pharmaceutical Sciences, Chulalongkorn University, Bangkok, Thailand

☯ These authors contributed equally to this work.
* chotima.boettcher@charite.de (CB); boonchoo.sr@chula.ac.th (BS)

**Data Availability Statement:** All relevant data are within the paper.

**Funding:** P.K. acknowledged the financial support of the scholarship from the Graduate School,

## Abstract

*Dendrobium* bibenzyls and phenanthrenes such as chrysotoxine, cypripedin, gigantol and moscatilin have been reported to show promising inhibitory effects on lung cancer growth and metastasis in *ex vivo* human cell line models, suggesting their potential for clinical application in patients with lung cancer. However, it remains to be determined whether these therapeutic effects can be also seen in primary human cells and/or *in vivo*. In this study, we comparatively investigated the immune modulatory effects of bibenzyls and phenanthrenes, including a novel *Dendrobium* bibenzyl derivative, in primary human monocytes. All compounds were isolated and purified from a Thai orchid *Dendrobium lindleyi* Steud, a new source of therapeutic compounds with promising potential of tissue culture production. We detected increased frequencies of TNF- and IL-6-expressing monocytes after treatment with gigantol and cypripedin, whereas chrysotoxine and moscatilin did not alter the expression of these cytokines in monocytes. Interestingly, the new 4,5-dihydroxy-3,3′,4′-trimethoxybibenzyl derivative showed dose-dependent immune modulatory effects in lipopolysaccharide (LPS)-treated CD14$^{lo}$ and CD14$^{hi}$ monocytes. Together, our findings show immune modulatory effects of the new bibenzyl derivative from *Dendrobium lindleyi* on different monocyte sub-populations. However, therapeutic consequences of these different monocyte populations on human diseases including cancer remain to be investigated.

## Introduction

The *Dendrobium* plants are vastly distributed in a large area including tropical and subtropical regions of Asia and North Australia [1]. Because of the extensive geographical distribution, *Dendrobium* becomes one of the largest genera in *Orchidaceae*, composing of approximately

Chulalongkorn University to commemorate the 72nd anniversary of his Majesty King Bhumibol Adulyadej. C.B. and J.P. were supported by the German Research Foundation (SFB TRR167, B05 & B07). J.P. received additional funding from the Berlin Institute of Health (CRG2aSP6) and the UK DRI (Momentum Award). B.S. is grateful to Faculty of Pharmaceutical Sciences, Chulalongkorn University for a research fund (Phar2563-RG010). The funders had no role in study design, data collection and analysis, decision to publish, or preparation of the manuscript.

**Competing interests:** The authors declare no competing financial interests.

1500–2000 species [2]. Of these, about 41 species have been used in traditional Chinese medicine [3]. Various groups of secondary metabolites have been reported, for instance, alkaloids, coumarins, bibenzyls, fluorenones, phenanthrenes, and sesquiterpenoids. Some of these metabolites exhibit a wide spectrum of pharmacological activities, including antioxidant, anti-inflammatory, antidiabetic, antitumor, antimicrobial and neuroprotective activities [2, 4]. For example, purified bibenzyls and phenanthrenes, such as chrysotoxine from *D. pulchellum*, cypripedin from *D. densiflorum*, gigantol from *D. draconis* and moscatilin from *D. brymerianum*, have been demonstrated to show promising inhibitory effects on human *in vitro* cell line models of lung cancer growth and metastasis [5–9]. However, whether these therapeutic effects can be also seen in primary human cells and/or *in vivo* remains to be investigated. Nevertheless, it is also unclear whether systemic administration of these compounds would influence phenotypes and/or functions of other cell types such as circulating immune cells in the peripheral blood.

The lung is a highly specialized organ that is continuously exposed to numerous pathogens, pollutants, oxidants, gases, and toxicants from the outside ambient air, which makes it susceptible to varying degrees of oxidative and inflammatory injury. Under some circumstances, inflammatory responses can result in the release of proinflammatory cytokines and chemokines, which in turn stimulate the influx of polymorphonuclear leukocytes (PMNs) and monocytes into the lung to combat the invading pathogens. However, chronic inflammation-induced cytokine production of monocytes in the lung may predispose individuals to lung cancer [10]. Tumor-infiltrating myeloid cells (TIMs), comprising monocytes, macrophages, dendritic cells and neutrophils, have emerged as key regulators of cancer growth and recurrence [11, 12]. The infiltration of monocytes into human specimens of non-small-cell lung cancer was found to be increased compared with normal lung tissue [12]. Although, it remains elusive, which monocyte sub-populations recruit into tumors, these infiltrating cells showed increased expression of chemokines and chemotactic activity in tumor environment [11]. Increased numbers and inflammatory activity of tumor-associated, monocyte-derived macrophages were linked to poor prognosis in lung cancer patients [12, 13]. Amelioration of monocyte-induced inflammation and/or infiltration may synergize to inhibit tumor growth and metastasis, and lead to a better prognosis of patients with lung cancer.

Furthermore, monocytes are crucial for the efficiency of the immune system, since they act in numerous immunological mechanisms, such as replenishment of resident macrophages, production of dendritic cell subsets, and acute and chronic inflammatory responses [14,15]. The cellular activity of these cells involves the production of several molecules responsible for killing of pathogens ($H_2O_2$, NO), cellular recruitment (IL-8, CCL2, and CCL3), pro-inflammatory activation (TNF, IL-1β, and IL-6), polarization of adaptive immune response (IL-12), regulation of inflammation (IL-10 and TGF-β1), and tissue repair (TGF-β1 and bFGF) [16]. Monocyte dysregulation, which is generally indicated by intense production of inflammatory mediators, has been observed in several inflammatory diseases, including sepsis [17], atherosclerosis [18], rheumatoid arthritis (RA) [19], and hepatic fibrosis [20]. High levels of TNF and IL-6 are associated with worse prognoses in septicemia [21] and progression and worsening of RA [22].

In this study, we aimed to investigate the immune modulatory activities on circulating monocytes of four known *Dendrobium* substances, which have already been shown to exert therapeutic effects in *in vitro* models of lung cancer [5–9], as well as one newly identified bibenzyl *Dendrobium* compound. All five compounds were isolated, purified and identified from *Dendrobium lindleyi* Steud, known in Thai as "Ueang Pueng" [23]. This orchid is widely distributed in India (Sikkim), China (Guangdong, Hainan, Guangxi, and Guizhou), Hongkong, Bhutan and Southeast Asia (Thailand, Myanmar, Laos, and Vietnam) [3]. To our

knowledge, chemical constituents and biological activities of compounds isolated from this orchid have never been investigated before. To assess their immune modulatory activities on circulating monocytes of the peripheral blood, we established an *ex vivo* model of primary human monocytes with low cell attachment, in which we attempted to minimize differentiation of monocyte into macrophage *ex vivo*. Our results revealed increased frequencies of TNF- and IL-6-expressing monocytes after treatment with gigantol and cypripedin, whereas chrysotoxine and moscatilin did not alter the expression of these cytokines in monocytes. Interestingly, the new 4,5-dihydroxy-3,3′,4′-trimethoxybibenzyl derivative alleviated lipopolysaccharide (LPS)-induced cytokine production of primary human monocytes, suggesting immune modulatory activity.

## Materials and methods

### Plant material

The whole plant of *D. lindleyi* was purchased in September 2009 from Chatuchak market, Bangkok, Thailand (13˚47'57.1"N, 100˚32'55.2"E). Plant identification was done by one of the authors (B. Sritularak). A voucher specimen (BS-DL-092552) has been deposited at the Department of Pharmacognosy and Pharmaceutical Botany, Faculty of Pharmaceutical Sciences, Chulalongkorn University.

### Extraction and isolation

*Dendrobium lindleyi* (1.2 kg) was extracted as previously described [26]. Briefly, the dried whole plant was grounded extracted with methanol (MeOH) at room temperature (RT). The extract was then evaporated under reduced pressure. The MeOH residue (97 g) was suspended in water and partitioned with immiscible organic solvents comprising EtOAc and *n*-BuOH. The EtOAc extract (34 g) was fractionated on silica gel column (EtOAc-hexane, gradient) to yield in total of 10 fractions (A-J). Fraction F (3.2 g) was separated by column chromatography over silica gel (EtOAc-hexane, gradient) to obtain 13 fractions (FI-FXIII). Fraction FVIII (140 mg) was purified on Sephadex LH-20 column (MeOH) to furnish chrysotoxine (#**2**) (90 mg), gigantol (#**3**) (20 mg) and cypripedin (#**4**) (25 mg), respectively. Moscatilin (#**5**) (90 mg) was purified from fraction FX (340 mg) on Sephadex LH-20 (MeOH). Fraction G (2.6 g) was fractionated by CC ((silica gel, EtOAc-hexane, gradient) and then by Sephadex LH-20 (MeOH) to afford #**1** (47 mg).

### Analytical procedures

Bruker micro TOF mass spectrometer (ESI-MS) was used for mass spectrometric analysis and a Milton Roy Spectronic 300 Array spectrophotometer was utilized for UV spectroscopic measurement. NMR spectra were obtained on a Bruker Avance III HD 500 NMR spectrometer or a Bruker Avance DPX-300 FT-NMR spectrometer. The recording of IR spectra was done by a Perkin-Elmer FT-IR 1760X spectrophotometer. Column chromatography (CC) was conducted on different stationary phases. These were silica gel 60 (Merck, Kieselgel 60, 70–320 µm), silica gel 60 (Merck, Kieselgel 60, 230–400 µm), C-18 (Merck, Kieselgel 60 RP-18, 40–63 µm) and Sephadex LH-20 (25–100 µm, GE Healthcare).

### Buffy coats and monocyte isolation

Buffy coats from healthy individuals were obtained from the German Red Cross (GRC). Blood donors gave informed consent to perform blood collection and to use the buffy coats samples for research. The research purpose was approved by the Ethics Committee of Charité –Universitätsmedizin Berlin.

Human peripheral blood mononuclear cells (PBMCs) were isolated and aliquoted at 20 x $10^6$ PBMCs (per mL) were cryopreserved in liquid nitrogen tank until analysis. Frozen PBMCs of three biologically independent donors were thawed, washed and pooled in MACS buffer (0.5% BSA in PBS containing 2 mM EDTA). Monocytes were isolated using negative selection, pan-monocyte Isolation Kit (Miltenyi Biotec, Bergisch Gladbach, Germany) according to manufacturer's specifications. Briefly, PBMCs were resuspended in 120 μL of MACS buffer. Thirty micro liters of FcR blocking reagent and 30 μL of biotin-antibody cocktail were added, mixed thoroughly and incubated at 4˚C. After 5 min incubation, 90 μL of MACS buffer and 60 μL of anti-biotin micro beads were added and incubated at 4˚C for 10 min. Stained ells were then washed with MACS buffer and pelleted (4˚C, 300 x g, 8 min). The pellet was then resuspended in 500 μL of MACS buffer and loaded onto the MACS column. The column was then washed twice with MACS buffer. The flow-through and washed fraction containing unlabeled monocytes was collected. Cell number and viability were determined by 0.2% trypan blue staining.

## Cell culture and stimulation

Isolated monocytes were resuspended in 1 mL of RPMI1640 (Biochrom GmbH, Berlin, Germany) containing 10% heat-inactivated fetal calf serum (FCS) (Sigma-Aldrich, St. Louis, USA), penicillin (100 U/mL; Biochrom GmbH, Berlin, Germany) and streptomycin (100μg/ mL; Biochrom GmbH, Berlin, Germany). Cell concentration was adjusted to ~ 2 x $10^6$ cells/ mL. About 2 x $10^5$ cells (per well) were transferred into ultra-low-attachment 96-well plate (Corning, New York, USA). Different concentrations of compounds (Fig 3A) were added to the cell culture. To inhibit protein transport from Golgi apparatus to the endoplasmic reticulum, monensin (5 μg/mL; BioLegend, San Diego, USA) was also added right after the application of *Dendrobium* compounds. After two hours incubation, cells were stimulated by adding 100 ng/mL of lipopolysaccharide (LPS) (Sigma-Aldrich, St. Louis, USA). Cells were cultured for 4 more hours or overnight (18 hours). Finally, cells were analysed by flow cytometry. Control conditions were naïve cells (without treatment of *Dendrobium* compounds and LPS) and unstimulated cells that were previously treated with the *Dendrobium* compounds.

## Flow cytometry

Monocytes were incubated for 10 min in FcR-blocking buffer (1:100; Miltenyi Biotec, Bergisch Gladbach, Germany) at 4˚C, to block unspecific binding of antibodies to Fc-receptors. Next, cells were incubated with conjugated antibodies detecting extracellular epitopes (i.e. CD3-PE-Cy7 / SK7 / BioLegend; CD335-PE / 9E2 / BioLegend; CD16-APC-Cy7 / 3G8 / BioLegend; CD11c-APC / 3.9 / BioLegend; HLA-DR-pacific blue / L243 / BioLegend and CD14-FITC / RMO52 / Beckman) diluted in staining buffer (0.5% BSA in PBS containing 2 mM EDTA) for 20 min at 4˚C. Cells were then washed with 1 mL of staining buffer, and further incubated in 200 μL of intracellular fixation/permeation buffer (eBioscience, California, USA) for 30 min at 4˚C. After washing with 1 mL of permeation buffer (eBioscience, California, USA), 50 μL of antibody cocktail detecting intracellular proteins (i.e. TNF-brilliant violet / Mab11 / BioLegend; IL-6-PE-Cy7 / MQ2-13A5 / BioLegend; CCL2-PE / 5D3-F7 / BioLegend and CD68-PerCP-Cy5.5 / Y1/82A / BioLegend) was added and further incubated for 30 min at 4˚C. Cells were then washed with staining buffer and pelleted at 300 x g, 4˚C for 10 min. Stained cells were finally resuspended in staining buffer. Cell viability was determined using Propidium Iodide (25 μg/mL; Thermo Fisher Scientific, Massachusetts, USA). Protein expression was detected on a 3-laser BD FACSCanto II machine (BD Biosciences, New Jersey, USA) with software BD DIVA version 8.1. FlowJo software version 10.1 (Ashland, OR, USA) was used to determine the phenotype on the basis of fluorescence intensity.

## Statistical analysis

Acquired FACS data were extracted in the Flow Cytometry Standard (*.fcs) format. Cell populations were gated in FlowJo (Becton, Dickinson and Company, New Jersey, USA). Frequency of cells positive for a given signal was extracted into GraphPad Prism 8 (GraphPad Software Inc., California, USA) and statistically analyzed by a one-way ANOVA or a multiple t-test (as indicated), with a confidence interval of 95% ($\alpha = 0.05$).

## Results

### Establishment of low-attachment primary human monocyte cultures for functional analysis

It has been demonstrated that *ex vivo* culture of human monocytes using classical polypropylene cell culture plates resulted in phenotypic and functional changes, such as an increased granularity and reduced transendothelial diapedesis function [24]. Therefore, we newly established an *ex vivo* culture system using an ultra-low attachment multiple well-plate (Fig 1A). First, we used peripheral blood mononuclear cells (PBMCs) to optimize the culture system (Fig 1A and 1B). We observed a better preservation of the cell composition in the culture system using RPMI-1640 medium compared with reduced serum medium OptiMEM (Fig 1C and 1D).

Next, we tested the LPS-induced inflammatory responses of monocytes under these culture conditions. We isolated monocytes (both $CD14^{lo}$ and $CD14^{hi}$ monocytes) from PBMCs using magnetic activated cell sorting (MACS) (Figs 1A and 2A). Isolated monocytes were then cultured in the established low-attachment culture system in the presence of 100 ng/mL LPS or PBS (control) for 6 hours. Compared to control, the cell composition of the monocyte population was changed after LPS stimulation. Namely, a strong reduction of $CD14^+CD16^+$ monocytes and a moderately decreased frequency of $CD14^{hi}$ monocytes were found (Fig 2B). As expected, increased expression of TNF was detected in all three monocyte populations after LPS stimulation, with the highest response in $CD14^{hi}$ inflammatory monocytes (Fig 2C and 2D). Furthermore, we found increased expression of IL-6 in $CD14^{hi}$ and $CD14^{lo}$ monocytes after LPS stimulation, whereas MCP-1 (or CCL2) and CD68 were unchanged under these conditions (Fig 2E and 2F).

### *Dendrobium* compound isolation, purification and identification

Phytochemical investigation from the whole plant of *Dendrobium lindleyi* (Fig 3A) revealed the isolation of a novel bibenzyl (Fig 3A, **#1**), along with four known compounds, which include chrysotoxine (Fig 3A, **#2**) [5, 6], gigantol (Fig 3A, **#3**) [8, 25–28], cypripedin (Fig 3A, **#4**) [7,29,30] and moscatilin (Fig 3A, **#5**) [9,27,31,32]. The structure of a new compound was elucidated through analysis of its spectroscopic data.

Compound **#1** was obtained as a brown amorphous solid; UV (MeOH) $\lambda_{max}$ (log ε) 220 (4.09), 229 (4.08), 285 (3.76) nm; IR (film) $\nu_{max}$: 3428, 2924, 1724, 1629, 1605, 1515, 1453, 1263, 1093, 1027 cm$^{-1}$; $^1$H NMR (500 MHz, CDCl$_3$) and $^{13}$C NMR (125 MHz, CDCl$_3$) spectral data, see Fig 3B; HR-ESI-MS: $m/z$ 327.1212 [M+Na]$^+$ (calcd. for $C_{17}H_{20}O_5Na$, 327.1208). The positive HR-ESI-MS showed an [M+Na]$^+$ ion at $m/z$ 327.1212 (calcd. for $C_{17}H_{20}O_5Na$, 327.1208), suggesting the molecular formula $C_{17}H_{20}O_5$. The absorption bands of the IR spectrum were defined as hydroxyl (3428 cm$^{-1}$), aromatic ring (2924, 1605 cm$^{-1}$) and methylene (1453 cm$^{-1}$) groups. The UV absorptions at 220, 229 and 285 nm were indicative of a bibenzyl nucleus [33]. This was confirmed by the presence of two pairs of methylene protons at δ 2.81 (H$_2$-α) and 2.84 (H$_2$-α′) ppm, which correlated to the carbon atoms at δ 38.0 (C-α) and 37.6

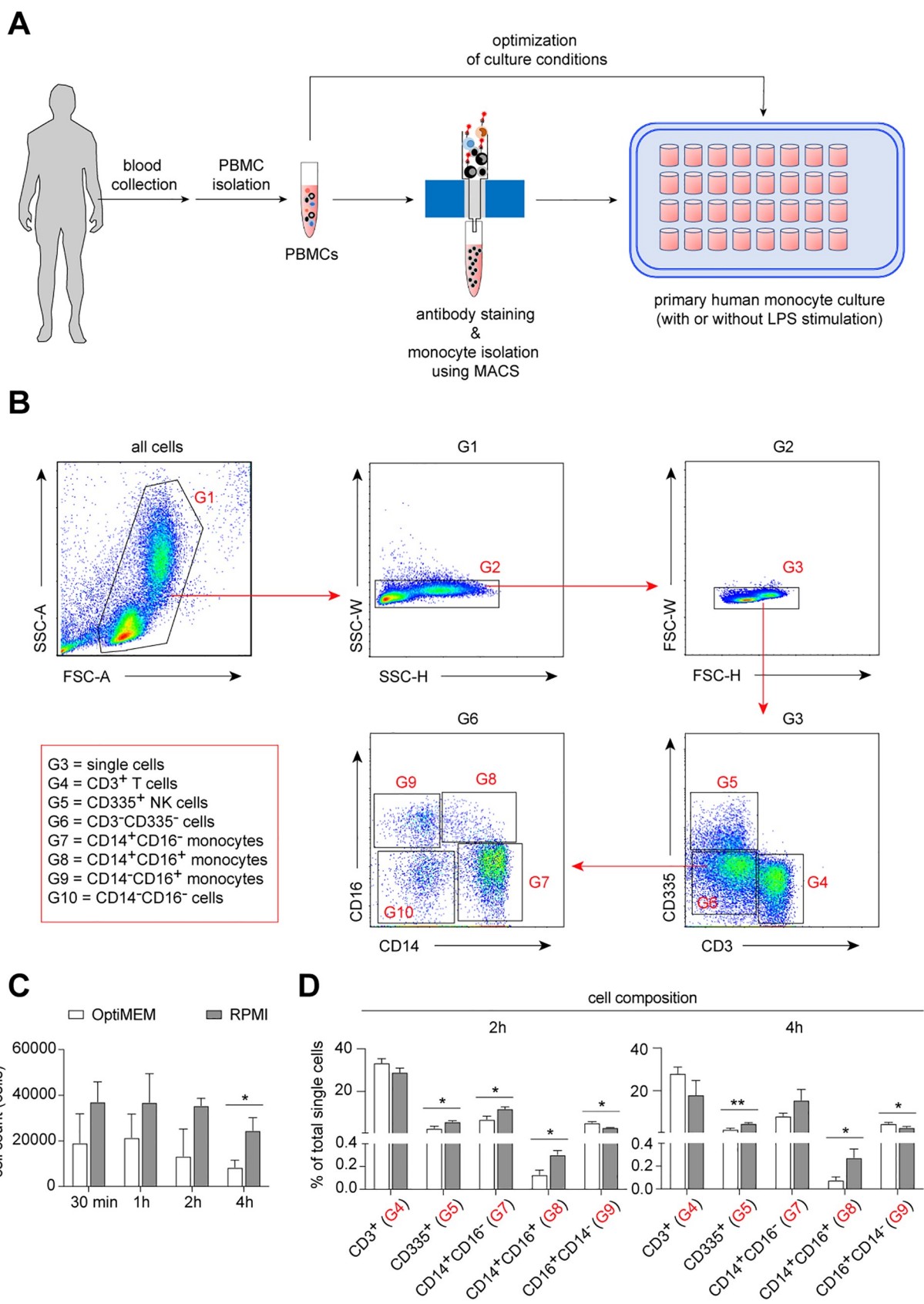

**Fig 1. Low-attached *in vitro* cultured model of human monocytes.** (**A**) Schematic representation of experimental procedure. (**B**) Dot plots demonstrate flow cytometric gating strategy used to obtain monocytes (G7, G8 and G9) for all samples. (**C**) Bar graphs show cell numbers in different culture conditions at different time points *in vitro*. (**D**) Bar graphs show different cell composition of PBMCs that were cultured in OptiMEM or RPMI medium conditions at 2 and 4 hours *in vitro*. Multiple t-test, corrected for multiple comparisons using Holm-Sidak method, $^*P < 0.05$, $^{**}P < 0.01$.

(C-$\alpha'$) ppm, respectively. The $^1$H NMR spectrum (Fig 3B) also displayed five aromatic protons at δ 6.26–6.81 ppm and three methoxyl groups at δ 3.85 (3H, s, MeO-3), 3.87 (3H, s, MeO-3′) and 3.88 (3H, s, MeO-4′). On ring A, the $^1$H NMR spectrum showed signals for two doublets at δ 6.26 ($J$ = 2.0 Hz, H-2) and 6.48 ($J$ = 2.0 Hz, H-6), which correlated to C-α in the HMBC spectrum. The first methoxyl group (δ 3.85) was located at C-3 based on its NOESY correlation with H-2. The $^1$H NMR ABM coupling system of ring B appeared at δ 6.69 (1H, d, $J$ = 2.0 Hz, H-2′), 6.73 (1H, dd, $J$ = 8.0, 2.0 Hz, H-6′) and 6.81 (1H, d, $J$ = 8.0 Hz, H-5′). The HMBC correlations of C-α′ with H-2′ and H-6′ and the NOESY cross-peak of two methoxy groups at δ 3.87 and δ 3.88 with H-2′ and H-5′, respectively, placed the second methoxyl group at C-3′ (δ 148.7) and the third methoxyl group at C-4′ (δ 147.2). Based on above spectral data, compound **#1** was identified as 4,5-dihydroxy-3,3′,4′-trimethoxybibenzyl.

## Evaluation of immune modulatory effects of *Dendrobium* compounds

To assess immune modulatory effects of all five *Dendrobium* compounds, we cultured MACS-isolated primary human monocytes under our established culture conditions (Figs 1 and 2), including an inhibitor of intracellular protein transport–monensin, in the presence of *Dendrobium* compounds, which were 4,5-dihydroxy-3,3′,4′-trimethoxybibenzyl (**#1**), chrysotoxine (**#2**), gigantol (**#3**), cypripedin (**#4**) and moscatilin (**#5**). Since these five compounds were diluted in dimethyl sulfoxide (DMSO), the same concentration of DMSO was added to the control culture without any compounds. We tested three known therapeutic concentrations [5–9], which were 5, 10 and 20 μM. After 2 hours of incubation, we stimulated monocytes by adding 100 ng/mL LPS and incubated for four more hours. We observed an increased frequency of TNF-expressing cells in gigantol (**#3**)- and cypripedin (**#4**)-treated monocytes without LPS stimulation, suggesting immune modulatory effects of these compounds on monocyte responses *ex vivo* (Fig 4A and 4B). After LPS stimulation, we detected increased frequencies of TNF- and IL-6-expressing monocytes, which were most pronounced in CD14$^{hi}$ inflammatory monocytes (Fig 4A and 4B), which is in line with the results shown in Fig 2. Although we observed decreased frequencies of LPS-induced TNF- and IL-6-expressing cells in monocytes treated with 4,5-dihydroxy-3,3′,4′-trimethoxybibenzyl (**#1**), chrysotoxine (**#2**) and moscatilin (**#5**), only the compound **#1** showed promising dose-dependent positive effects (Fig 4A–4D). Therefore, we selected the compound **#1** for a further evaluation. We tested whether the immune modulatory effects of *Dendrobium* compounds on monocytes persisted also after long-term exposure to LPS. To do so, the isolated monocytes were first incubated in the presence of the compound **#1** for two hours followed by overnight LPS-stimulation in a presence of monensin (5 μg/mL). We observed decreased expression of CD14 of the overnight cultured monocytes, and an increased abundance of TNF- and IL-6-expressing CD11c$^{lo}$ monocytes (Fig 5A), especially in the CD14$^{hi}$HLA-DR$^+$ sub-population. Overnight exposure of monocytes to compound **#1** resulted in changes in the monocyte composition, indicating as an increased abundance of CD14$^{lo}$HLA-DR$^-$ and CD14$^{lo}$HLA-DR$^+$ cells and a decreased abundance of CD14$^{hi}$HLA-DR$^+$ cells (Fig 5B). These changes were found at a highest degree in monocytes treated with 20 μM of **#1**. Furthermore, these observed changes were more pronounced in LPS-stimulated monocytes (Fig 5B, +LPS). Similar to the short-term incubation (Fig 4), CD14$^{hi}$ monocytes were the main source of TNF- and IL-6-expressing cells (Fig 5A, 5C & 5D).

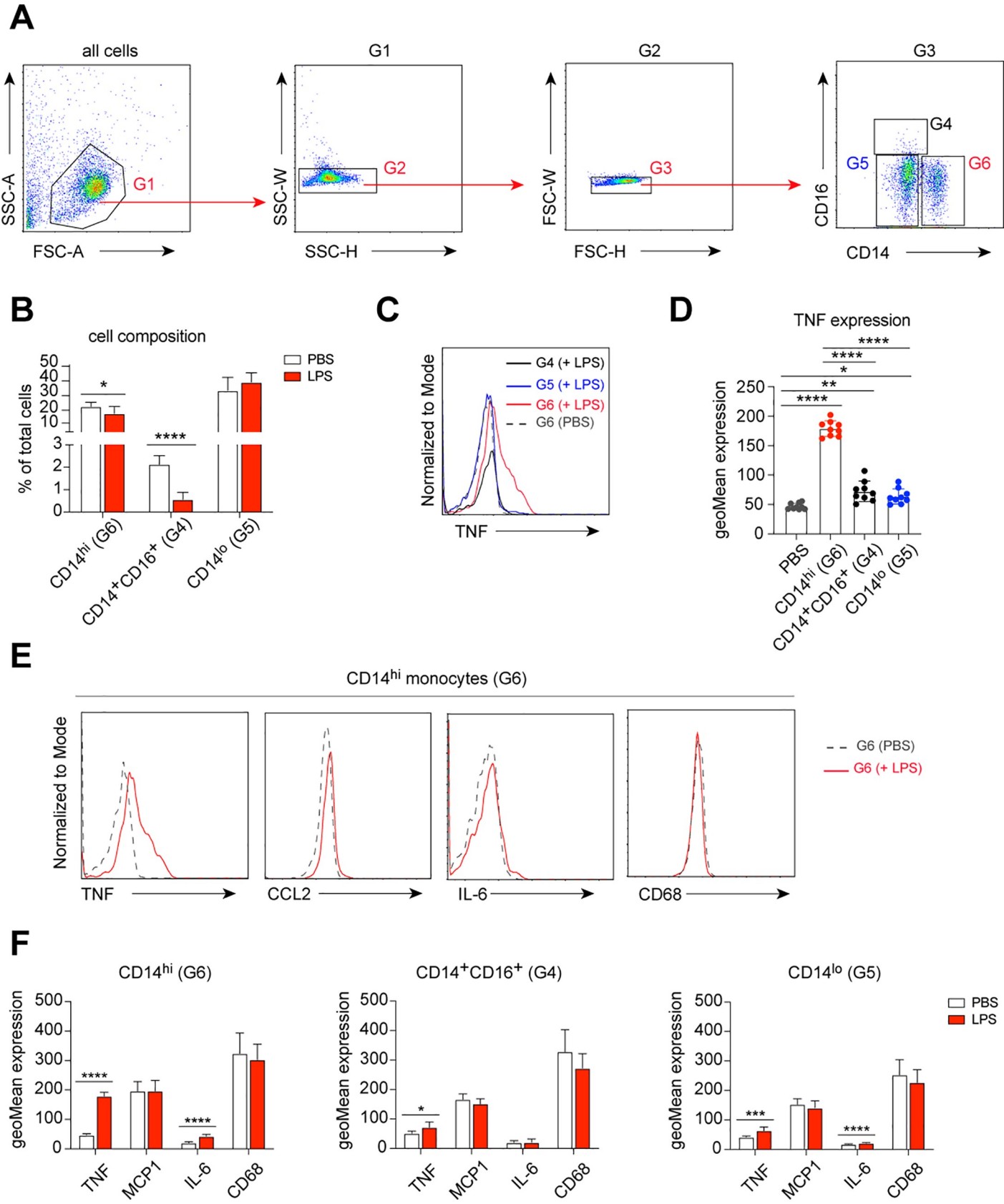

**Fig 2. LPS-induced cytokine production of cultured monocytes.** (**A**) Dot plots demonstrate flow cytometric gating strategy used to obtain monocytes (G4, G5 and G6) for all samples. (**B**) Bar graphs show changes in monocyte composition after LPS stimulation, compared with naïve monocytes. Multiple t-test, corrected for multiple comparisons using Holm-Sidak method, $^*P < 0.05$, $^{**}P < 0.01$, $^{***}P < 0.001$, $^{****}P < 0.0001$. (**C**) Histograms demonstrate fluorescent intensity representing LPS-induced TNF expression in different monocyte subsets, compared to naïve monocytes (unstimulated, PBS-treated). (**D**) Bar graphs show quantitative analysis of fluorescent intensity representing LPS-induced TNF expression in different monocyte subsets, compared to naïve monocytes (unstimulated, PBS-treated). Each dot represents an independent measurement. One-way ANOVA, corrected for multiple comparisons using Tukey Test, $^*P < 0.05$, $^{**}P < 0.01$, $^{***}P < 0.001$, $^{****}P < 0.0001$. (**E**) Histograms demonstrate fluorescent intensity representing LPS-induced cytokine expression of CD14$^{hi}$ monocytes (red line), compared to naïve CD14$^{hi}$ monocytes (unstimulated, PBS-treated; dashed line). (**F**) Bar graphs show quantitative analysis of fluorescent intensity representing LPS-induced cytokine expression in different monocyte subsets, compared to naïve monocytes (unstimulated, PBS-treated). One-way ANOVA, corrected for multiple comparisons using Tukey Test, $^*P < 0.05$, $^{**}P < 0.01$, $^{***}P < 0.001$, $^{****}P < 0.0001$.

Overnight incubation with 20 μM of the compound **#1** alone induced TNF expression in CD14$^{lo}$HLA-DR$^+$ and CD14$^{lo}$HLA-DR$^-$ cells (Fig 5C), while under LPS-stimulation it was downmodulating the LPS-induced TNF-expression (Fig 5C). In comparison to the control cells (without the compound **#1**), a treatment with 10 μM of **#1** could decreased LPS-induced TNF expression in all monocyte subsets (Fig 5C). The treatments with either 5 or 20 μM of **#1** could only decreased the LPS-induced TNF expression in CD14$^{hi}$HLA-DR$^+$ monocytes (Fig 5C). Without LPS-stimulation, the compound **#1** induced IL-6 expression in both CD14$^{lo}$ and CD14$^{hi}$HLA-DR$^+$ monocytes, but not in CD14$^{lo}$HLA-DR$^-$ cells (Fig 5D). Interestingly, the 5 and 10 μM of the compound **#1** strongly induced IL-6 expression in all monocyte subsets under LPS stimulation conditions, whereas the 20 μM concentration did not affect the expression of IL-6 in all three subsets (Fig 5D). The findings demonstrate a dose-dependent activating and/or immune modulatory effects of the compound **#1**, which are observed to be different in different monocyte sub-populations.

## Discussion

In this study, we demonstrated the use of low-attached culture model of primary human monocytes as a tool to study immune modulatory effects of purified natural products (the *Dendrobium* compounds). We showed the immune modulatory potential of a novel *Dendrobium* compound: 4,5-dihydroxy-3,3′,4′-trimethoxybibenzyl, which appeared to be dose-dependent and different between monocyte subsets. In addition, we have tested those known compounds including chrysotoxine (**#2**), gigantol (**#3**), cypripedin (**#4**) and moscatilin (**#5**), which were proposed as potential candidates for the development of cancer therapy. However, these compounds appeared to have either less immune modulatory activity or even more inflammatory effect on primary monocytes, compared with the compound **#1**. Our findings suggest an importance and/or feasibility of using *ex vivo* studied models of different primary human (immune) cells to estimate the therapeutic potential and/or any potential adverse effects of purified natural products, which will further facilitate the adjustment of developmental potential of each isolated compound in diseases.

In the low-attached monocyte culture model used in this study, we found stronger responses to TLR4 agonist LPS of CD14$^{hi}$ monocytes, compared to CD14$^{lo}$ monocytes, on the basis of the expression of inflammatory cytokines especially TNF and IL-6. Our results are in line with the study by Cros *et al.* in which CD14$^{dim}$ was demonstrated to respond poorly to agonists of TLR1, TLR2 and TLR4 [34]. The inflammatory cytokine TNF has been shown to be involved in many biological processes including fever, apoptosis, and infection-induced cachexia [16], as well as in inflammation-associated diseases, for example rheumatoid arthritis [35], acquired generalized lipodystrophy and combined Crohn's disease [36], Crohn's disease [37], and type 2 diabetes [38]. Moreover, chemokine such as TNF can also induce activate inflammatory responses, and are implicated in the regulation of tumor development and growth via regulation of tumor-associated angiogenesis, by activation of host immunological

## A

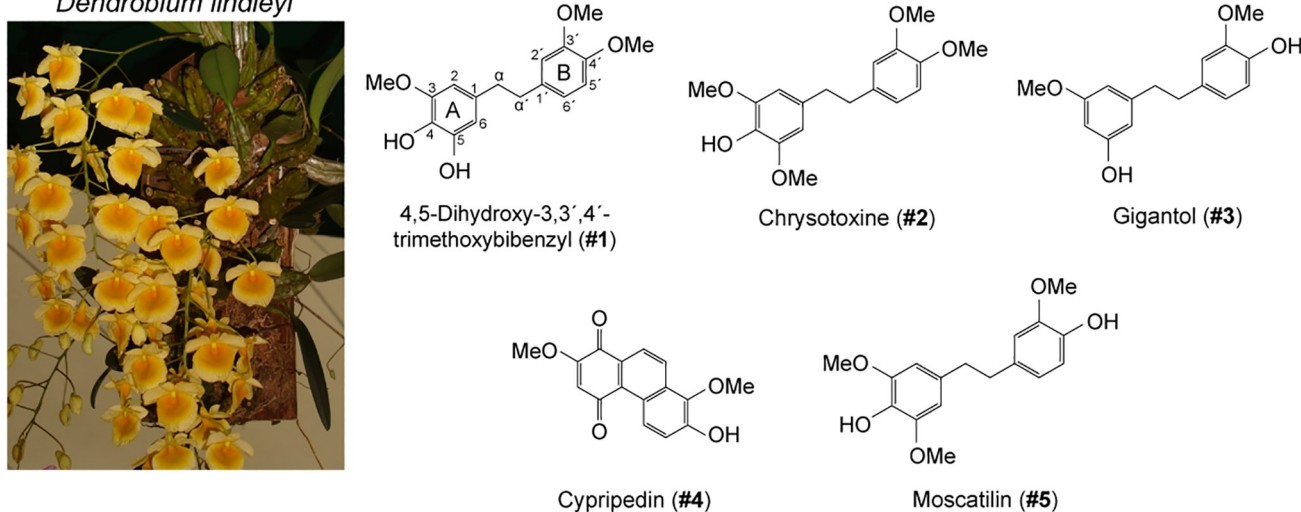

## B

### $^1$H NMR (500 MHz) and $^{13}$C NMR (125 MHz) spectral data of #1 in CDCl3

| Position | $^1$H | $^{13}$C | HMBC (correlation with $^1$H) |
|---|---|---|---|
| 1 | - | 133.7 | α*, α´ |
| 2 | 6.26 (d, $J$ = 2.0 Hz) | 103.5 | 6, α |
| 3 | - | 146.6 | 3-OMe, 2*, 4-OH |
| 4 | - | 130.5 | 2,6,4-OH*, 5-OH |
| 5 | - | 143.7 | 6*, 4-OH, 5-OH* |
| 6 | 6.48 (d, $J$ = 2.0 Hz) | 108.6 | 2, α, 5-OH |
| 1´ | - | 134.4 | α, α´*, 5´ |
| 2´ | 6.69 (d, $J$ = 2.0 Hz) | 111.9 | 6´, α´ |
| 3´ | - | 148.7 | 5´, 3´-OMe |
| 4´ | - | 147.2 | 2´, 6´, 4´-OMe |
| 5´ | 6.81 (d, $J$ = 8.0 Hz) | 111.2 | - |
| 6´ | 6.73 (dd, $J$ = 8.0, 2.0 Hz) | 120.3 | 2´, α´ |
| α | 2.81 (m) | 38.0 | α´*, 2, 6 |
| α´ | 2.84 (m) | 37.6 | α*, 2´, 6´ |
| MeOH-3 | 3.85 (s) | 56.1 | - |
| MeOH-3´ | 3.87 (s) | 55.9 | - |
| MeOH-4´ | 3.88 (s) | 55.8 | - |
| HO-4 | 5.25 (s) | - | - |
| HO-5 | 5.27 (s) | - | - |

*Two-bond coupling

HMBC denotes heteronuclear multiple bond correlation

**Fig 3. Isolated compounds from *Dendrobium lindleyi*.** (**A**) Chemical structures of five compounds isolated and purified from *Dendrobium lindleyi*. (**B**) Chemical structure characterization of 4,5-dihydroxy-3,3′,4′-trimethoxybibenzyl using $^1$H NMR and $^{13}$C NMR.

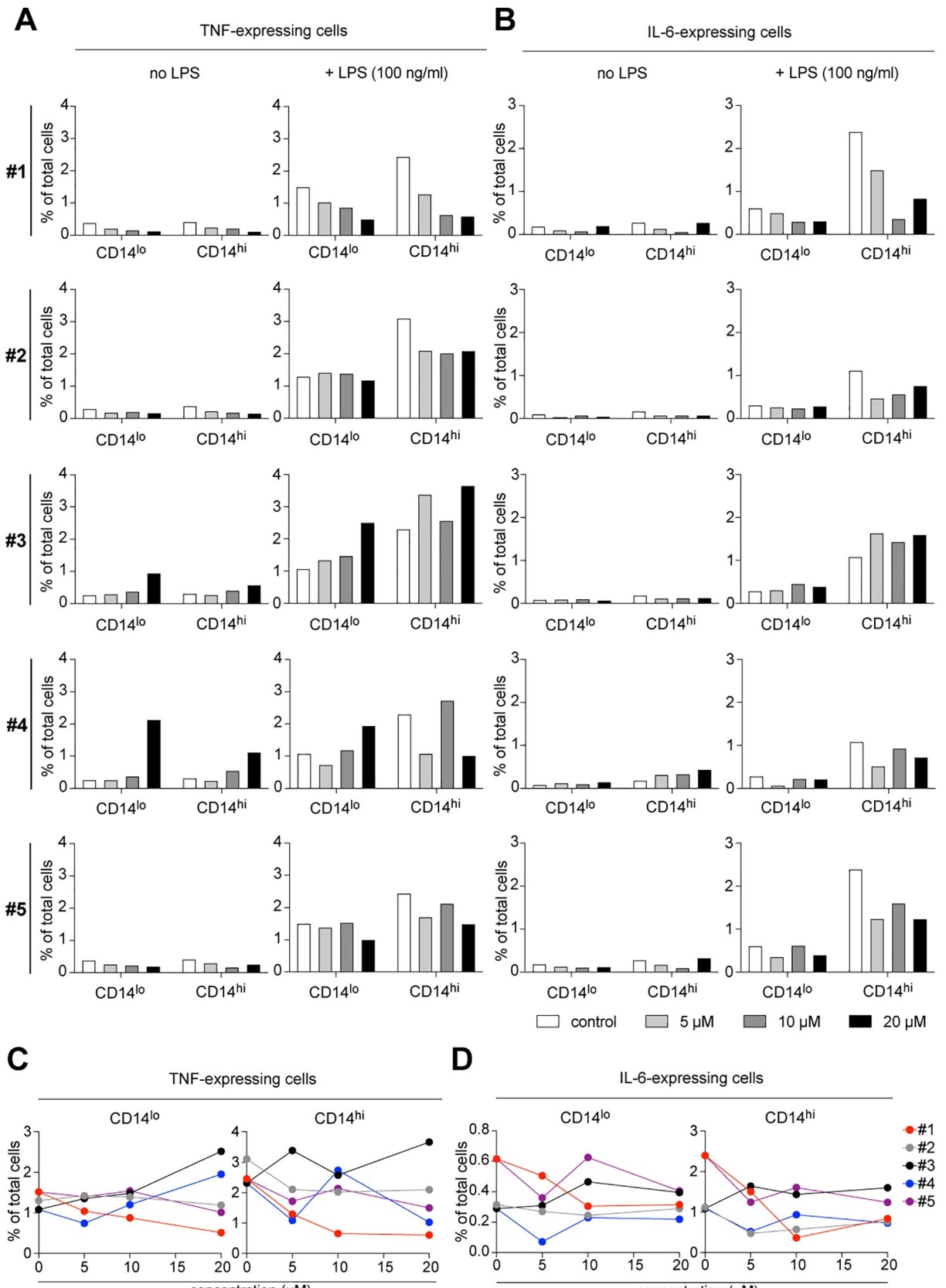

**Fig 4. Immune modulatory effect of isolated compounds from *Dendrobium lindleyi* on LPS-stimulated human monocytes.** Immune modulatory effect of all five *Dendrobium* compounds on LPS-induced TNF (**A,C**) and IL-6 (**B,D**) expression of monocytes *in vitro* were evaluated. (**A-B**) Bar graphs show the mean frequency (%) of CD14^lo and CD14^hi monocytes, which express TNF and IL-6, compared to naïve monocytes (no LPS). This is an explorative screening experiment, in which each condition (or each concentration) was analyzed in duplicate, thus no variation was shown. (**C-D**) The X-Y graphs show the correlation between the frequency (%) of TNF- or IL-6-expressing CD14^hi or CD14^lo monocytes and the concentration of all five *Dendrobium* compounds (#1 - #5).

responses or by direct inhibition of tumor cell proliferation [39]. It has been also shown that suppressing the increased expression of inflammation-related factors such as TNF is considered to potentially suppress the migration of macrophages into tumor tissues and regulate inflammatory changes in the tumor environments, respectively [40]. In this study, we detected *Dendrobium* compound-induced TNF expression in monocytes treated with 20 µM of gigantol (**#3**) or cypripedin (**#4**), suggesting potentially inflammatory effects of these compounds at high concentration. Whereas 4,5-dihydroxy-3,3′,4′-trimethoxybibenzyl (**#1**), chrysotoxine (**#2**) and moscatilin (**#5**) did not show obvious inflammatory effects on monocytes *in vitro*, only the novel compound **#1** has shown promising immune modulatory effects in a dose-dependent manner. However, in a long-term LPS-stimulation, only the 10 µM of the compound **#1** could downmodulate the LPS-induced TNF expression in all monocyte subsets, whereas the concentration of 20 µM induced TNF expression of unstimulated cells, suggesting dose- and treatment-duration-dependent immune modulatory effects and/or inflammatory effects of this compound.

Interleukin-6 (IL-6) is a cytokine with pleiotropic function, which are involved in host defense. In acute infection and/or tissue injury, monocytes/macrophages promptly produce IL-6 and contribute to removal of infectious agents and regeneration of damaged tissue through activation of immune, hematological, and acute-phase responses [41]. However, a dysregulation of IL-6 production and/or persistently increased IL-6 expression plays a pathological role in the development of various inflammatory diseases and cancers, thus IL-6 is a double-edged sword for the host [41]. Thus, the proper IL-6 expression is very important for host defense. In this study, although we did observe the reduction of the LPS-induced IL-6 production in monocytes, that were short-term treated with the compound **#1**, we found strongly increased production of IL-6 in long-term treated monocytes (Fig 5). However, since we detected a decrease in LPS-induced TNF-expression under the same conditions, an increased IL-6 expression observed in this study may refer to increased monocyte activity against LPS-induced inflammation.

## Conclusions

In summary, we demonstrate herein immune modulatory effects of the new compound 4,5-dihydroxy-3,3′,4′-trimethoxybibenzyl on different monocyte subsets *ex vivo*, which can potentially be developed as *in vivo* immune modulatory drug in a broad spectrum of inflammation-driven diseases besides lung carcinomas. However, since *Dendrobium lindleyi* is widely distributed, using this orchid for drug development may have a risk of habitat destruction, over-collection and commercially trade due to low natural regeneration rates. Moreover, *Dendrobium* plants are listed on Appendix II (*D. cruentum* is the only species listed on Appendix I) of Convention on International Trade in Endangered Species of Wild Fauna and Flora [42], which monitors, regulates or bans the trade to ensure species survival [43]. Therefore, in parallel to drug development, an establishment and improvement of methodologies/techniques for seedling propagation and artificial cultivation technology such as tissue culture and artificial-sheltered cultivation [44–47] are required to ensure sustainable development and future marketing.

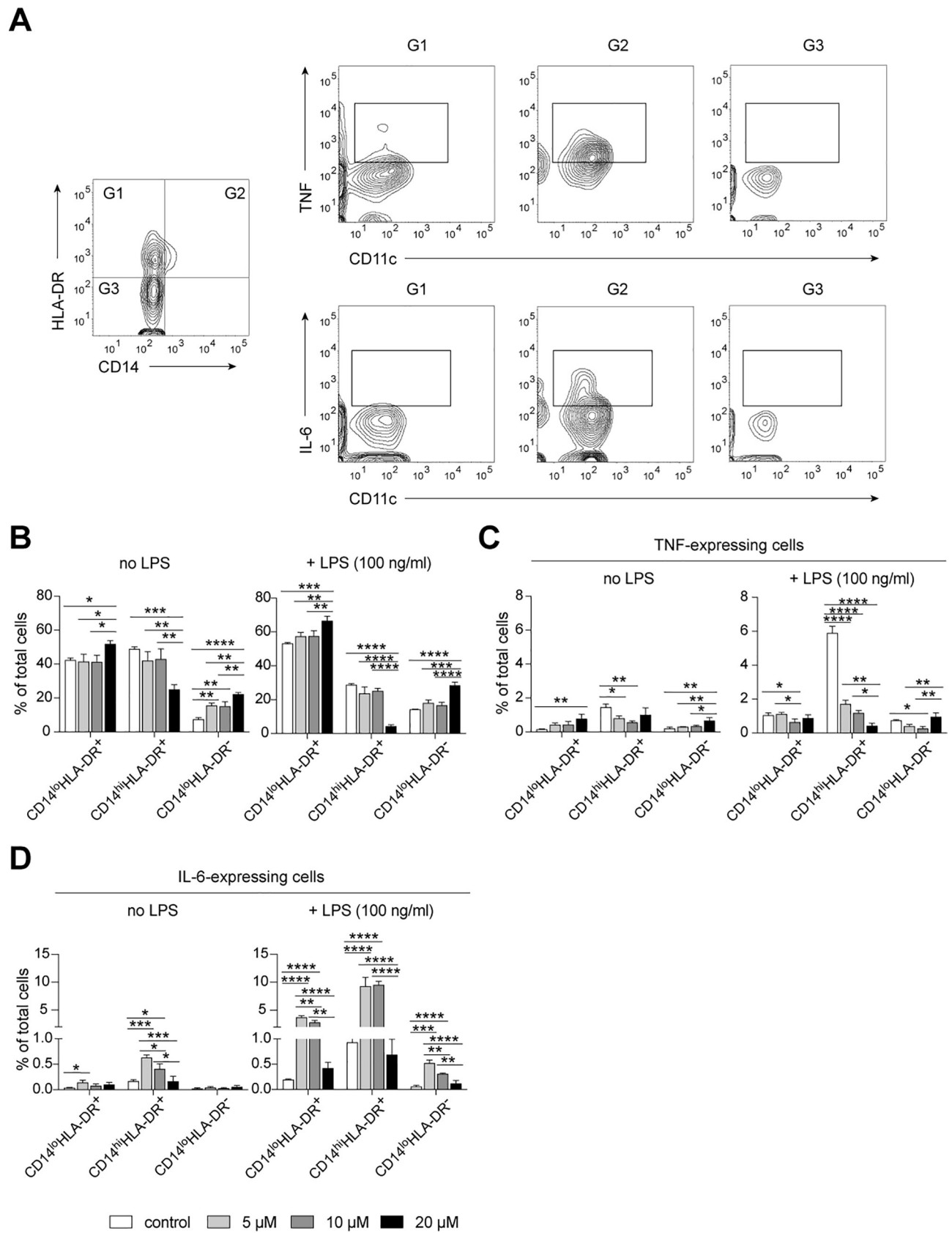

**Fig 5. Immune modulatory effect of 4,5-dihydroxy-3,3′,4′-trimethoxybibenzyl on long-term LPS-stimulation of human monocytes.** (**A**) Contour plots demonstrate flow cytometric gating strategy used to analyze the long-term culture of low-attached monocytes for all samples. Expression of TNF and IL-6 of three monocyte subsets (G1, G2 and G3) were shown. (**B**) Bar graphs show changes in monocyte composition after overnight incubation with 5, 10 or 20 μM of 4,5-dihydroxy-3,3′,4′-trimethoxybibenzyl with (+ LPS) or without LPS (no LPS) stimulation. (**C**) Bar graphs show the mean frequency (%) of TNF-expressing monocytes after overnight incubation with 5, 10 or 20 μM of 4,5-dihydroxy-3,3',4'-trimethoxybibenzyl with (+ LPS) or without LPS (no LPS) stimulation. (**D**) Bar graphs show the mean frequency (%) of IL-6-expressing monocytes after overnight incubation with 5, 10 or 20 μM of 4,5-dihydroxy-3,3′,4′-trimethoxybibenzyl with (+ LPS) or without LPS (no LPS) stimulation. One-way ANOVA, corrected for multiple comparisons using Tukey Test, $^*P < 0.05$, $^{**}P < 0.01$, $^{***}P < 0.001$, $^{****}P < 0.0001$.

## Acknowledgments

The authors are grateful to Christian Böttcher for providing helpful technical support with the generation of PBMCs.

## Author Contributions

**Conceptualization:** Pichayatri Khoonrit, Josef Priller, Chotima Böttcher, Boonchoo Sritularak.

**Data curation:** Pichayatri Khoonrit, Alp Mirdogan, Adeline Dehlinger, Kittisak Likhitwitaya-wuid, Chotima Böttcher, Boonchoo Sritularak.

**Formal analysis:** Chotima Böttcher, Boonchoo Sritularak.

**Funding acquisition:** Josef Priller, Chotima Böttcher, Boonchoo Sritularak.

**Investigation:** Chotima Böttcher, Boonchoo Sritularak.

**Methodology:** Pichayatri Khoonrit, Alp Mirdogan, Adeline Dehlinger, Wanwimon Mekboon-songlarp, Kittisak Likhitwitayawuid, Chotima Böttcher, Boonchoo Sritularak.

**Project administration:** Boonchoo Sritularak.

**Validation:** Pichayatri Khoonrit, Alp Mirdogan, Chotima Böttcher, Boonchoo Sritularak.

**Visualization:** Pichayatri Khoonrit, Alp Mirdogan, Chotima Böttcher, Boonchoo Sritularak.

**Writing – original draft:** Chotima Böttcher, Boonchoo Sritularak.

**Writing – review & editing:** Josef Priller, Chotima Böttcher, Boonchoo Sritularak.

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
