## [Decision Letter · Decision Letter 0]

17 Jun 2020

PONE-D-20-02679

Immune modulatory effect of a novel 4,5-dihydroxy-3,3´,4´-trimethoxybibenzyl from Dendrobium lindleyi

PLOS ONE

Dear Dr. Böttcher,

Thank you for submitting your manuscript to PLOS ONE. After careful consideration, we feel that it has merit but does not fully meet PLOS ONE’s publication criteria as it currently stands. Therefore, we invite you to submit a revised version of the manuscript that addresses the points raised during the review process.

Your manuscript has been assessed by two reviewers, considered experts in the field. According to their suggestion, some experimental data should be shown. Please address their comments in a point-by-point answer to the reviewers.

We look forward to receiving your revised manuscript.

Kind regards,

Horacio Bach

Academic Editor

PLOS ONE

Journal Requirements:

2. We note that Figures in your submission contain copyrighted images.

All PLOS content is published under the Creative Commons Attribution License (CC BY 4.0), which means that the manuscript, images, and Supporting Information files will be freely available online, and any third party is permitted to access, download, copy, distribute, and use these materials in any way, even commercially, with proper attribution. For more information, see our copyright guidelines: http://journals.plos.org/plosone/s/licenses-and-copyright.

a.         You may seek permission from the original copyright holder of Figure(s) [#] to publish the content specifically under the CC BY 4.0 license.

3. We noticed you have some minor occurrence(s) of overlapping text with the following previous publication(s), which needs to be addressed:

https://doi.org/10.1080/14786419.2018.1527839

https://doi.org/10.1371/journal.pone.0198682

In your revision ensure you cite all your sources (including your own works), and quote or rephrase any duplicated text outside the Methods section.

Further consideration is dependent on these concerns being addressed.

4. Please provide additional details regarding participant consent.

In the ethics statement in the Methods and online submission information, please ensure that you have specified (1) whether consent was suitably informed and (2) what type you obtained (for instance, written or verbal).

If your study included minors under age 18, state whether you obtained consent from parents or guardians.

If the need for consent was waived by the ethics committee, please include this information.

5. Please include in the methods the exact geographic location of the market from which you purchased D. lindleyi.

6. Your ethics statement must appear in the Methods section of your manuscript. If your ethics statement is written in any section besides the Methods, please move it to the Methods section and delete it from any other section. Please also ensure that your ethics statement is included in your manuscript, as the ethics section of your online submission will not be published alongside your manuscript.

Reviewers' comments:

Reviewer's Responses to Questions

**Comments to the Author**

1. Is the manuscript technically sound, and do the data support the conclusions?

Reviewer #1: Yes

Reviewer #2: Partly

2. Has the statistical analysis been performed appropriately and rigorously? 

Reviewer #1: Yes

Reviewer #2: Yes

3. Have the authors made all data underlying the findings in their manuscript fully available?

Reviewer #1: Yes

Reviewer #2: No

4. Is the manuscript presented in an intelligible fashion and written in standard English?

Reviewer #1: Yes

Reviewer #2: Yes

5. Review Comments to the Author

Reviewer #1: The manuscript of Pichayatri Khoonrit et al. describes the potential immunomodulatory properties of compounds isolated from Thai orchid Dendrobium lindleyi. The manuscript is significant as the presented study determine the first report of chemical constituents and biological activities of gigantol, cypripedin, chrysotoxine, moscatilin and newly discovered ,4,5-dihydroxy-3,3’,4’-trimethoxybibenzyl derivative, isolated form the orchid Dendrobium lindleyi. In order to assess the immune modulatory activities of these compounds, the authors established an ex vivo culture model of primary human monocytes with low cell attachment, which allows to avoid the phenotypic and functional changes, such as an increased granularity and reduced transendothelial diapedesis function in monocytes. The results of the study revealed an increased frequencies of TNF- and IL-6-expressing monocytes after cells treatment with two of five compounds (gigantol and cypripedin), whereas chrysotoxine and moscatilin did not upregulated the expression of these cytokines in monocytes. Moreover, the authors showed that the newly isolated 4,5-dihydroxy-3,3˘,4˘-trimethoxybibenzyl derivative downregulated lipopolysaccharide-induced cytokine production of primary human monocytes and concluded that these results suggest an anti-inflammatory activity of this agent. The study is preliminary, however it provides novel information regarding the immunomodulatory effects of orchid-derived compounds, which were previously described as promising inhibitory agents of lung cancer growth and metastasis. The flow cytometry experiments are well-planned and the results are clearly presented. However, I have some suggestions to the authors.

Concern #1: The manuscript is well written, however some typographical and grammatical errors have appeared:

- Page 4, line 2 – ‘…which have been already been shown..’, the second ‘been’ has to be removed.

- Page 6, Line 21 - The sentence: „To do so, the isolated monocytes were first incubated in the presence of the compound #1 for two hours followed by overnight LPS-stimulation in a presence of monensin (5 μg/ml) longer incubation times with the Dendrobium compounds” has to be corrected.

Concern #2: Page 5, line 4 – I recommend to add two abbreviations of monocyte chemoattractant protein-1 (MCP-1/CCL2), since on figure 2, both of these shortcuts appeared (Fig. 2E – CCL2; Fig. 2F – MCP-1).

Concern #3: In my opinion the figures 4C and 4D can be deleted since they display the same data as figures 4A and 4B.

Concern #4: The Introduction section is ended with the sentence :’Interestingly, the new 4,5-dihydroxy-3,3’,4’-trimethoxybibenzyl derivative alleviated lipopolysaccharide (LPS)-induced cytokine production of primary human monocytes, suggesting anti-inflammatory activity.’ This summary is too far-reaching, therefore the information about further validation of the results has to be added (examples: However, these results need to be further validated in in vivo studies or further studies elucidating the molecular pathway are needed).

Concern #5: The results represent well designed flow cytometry study, however, the measured parameters are limited. In the future, it would have been further interesting if the effect of examined compounds, most of all, the new 4,5-dihydroxy-3,3˘,4˘-trimethoxybibenzyl derivative, will be checked on other parameters, like the production of molecules responsible for killing of pathogens (H2O2, NO) or the phagocytic ability of human monocytes.

Concern #6: The authors order is different in “Orders of Authors” section and in the manuscript text.

Reviewer #2: The fact is that macrophages, an important population of cells, are involved primarily in defense against infectious disease and play a central role in maintaining all tissues in a healthy state.On the other hand, macrophages also have a multifaceted role in diseases such as cancer, Alzheimer disease, multiple sclerosis and type 1 diabetes… I think that the topic is interesting, the introduction is sufficient to understand the issue, but that the results within the text should be confirmed by statistics. The methods are well described as well as the presentation of the obtained results (Figures). However, I would like to see more data confirming the macrophage function after treatment with isolated components from a Thai orchid Dendrobium lindleyi Steud such as phagocytosis ability, confirmation of cytotoxicity in co-culture with tumor cells and measurement of NO and H2O2 levels as well as confirmation of anti-inflammatory effect of components through COX-2 inhibition and LOX activities.I believe that confirming the functional ability of macrophages could better link the importance of secretion of individual cytokines with respect to cytokine pleiotropy and redundancy and better and more clearly confirm their immunomodulatory role in health or disease. By confirming the functional capacity of macrophages, the whole research would confirm the greater possibility of application of these components and better justify the mentioned project in plant protection. Such a presentation gives only possible assumptions without confirming the true immunomodulatory role of the isolated components.

Second, minor problems, there are some mistakes about editorial handling, e.g. % must be written close the number. Please harmonize the way of writing degree C. SI units should be used… (e.g. μg/ml should be μg/mL). Please separate the SI unit from the number. In vitro and in vivo should be written in italic...... . Statistical value P should be written in italic and with a capital letter. Abbreviations must be explained the first time they are used, both in the Abstract and again in the main text.

6. PLOS authors have the option to publish the peer review history of their article (what does this mean?). If published, this will include your full peer review and any attached files.

Reviewer #1: No

Reviewer #2: No

---

## [Author Response · Author response to Decision Letter 0]

23 Jul 2020

Reviewer #1: 

The manuscript of Pichayatri Khoonrit et al. describes the potential immunomodulatory properties of compounds isolated from Thai orchid Dendrobium lindleyi. The manuscript is significant as the presented study determine the first report of chemical constituents and biological activities of gigantol, cypripedin, chrysotoxine, moscatilin and newly discovered ,4,5-dihydroxy-3,3’,4’-trimethoxybibenzyl derivative, isolated form the orchid Dendrobium lindleyi. In order to assess the immune modulatory activities of these compounds, the authors established an ex vivo culture model of primary human monocytes with low cell attachment, which allows to avoid the phenotypic and functional changes, such as an increased granularity and reduced transendothelial diapedesis function in monocytes. The results of the study revealed an increased frequencies of TNF- and IL-6-expressing monocytes after cells treatment with two of five compounds (gigantol and cypripedin), whereas chrysotoxine and moscatilin did not upregulated the expression of these cytokines in monocytes. Moreover, the authors showed that the newly isolated 4,5-dihydroxy-3,3˘,4˘-trimethoxybibenzyl derivative downregulated lipopolysaccharide-induced cytokine production of primary human monocytes and concluded that these results suggest an anti-inflammatory activity of this agent. The study is preliminary, however it provides novel information regarding the immunomodulatory effects of orchid-derived compounds, which were previously described as promising inhibitory agents of lung cancer growth and metastasis. The flow cytometry experiments are well-planned and the results are clearly presented. 

Response:

We thank the reviewer for this very supportive comment.

However, I have some suggestions to the authors.

Concern #1: The manuscript is well written, however some typographical and grammatical errors have appeared:

- Page 4, line 2 – ‘…which have been already been shown..’, the second ‘been’ has to be removed.

- Page 6, Line 21 - The sentence: „To do so, the isolated monocytes were first incubated in the presence of the compound #1 for two hours followed by overnight LPS-stimulation in a presence of monensin (5 μg/ml) longer incubation times with the Dendrobium compounds” has to be corrected.

Response:

We sincerely thank the reviewer for thorough proof-reading and apologize for such a mistake. We have corrected these sentences as suggested.

Concern #2: Page 5, line 4 – I recommend to add two abbreviations of monocyte chemoattractant protein-1 (MCP-1/CCL2), since on figure 2, both of these shortcuts appeared (Fig. 2E – CCL2; Fig. 2F – MCP-1).

Response:

We thank the reviewer for this suggestion and have revised the text accordingly.

Concern #3: In my opinion the figures 4C and 4D can be deleted since they display the same data as figures 4A and 4B.

Response:

We thank the reviewer for the comment. However, although they are a kind of redundancy, we think that the Fig 4C and 4D would serve as a good summary of Fig 4A and 4B, which would help readers to directly compare all conditions together. Therefore, we would like to keep the Fig. 4C and 4D.

Concern #4: The Introduction section is ended with the sentence :’Interestingly, the new 4,5-dihydroxy-3,3’,4’-trimethoxybibenzyl derivative alleviated lipopolysaccharide (LPS)-induced cytokine production of primary human monocytes, suggesting anti-inflammatory activity.’ This summary is too far-reaching, therefore the information about further validation of the results has to be added (examples: However, these results need to be further validated in in vivo studies or further studies elucidating the molecular pathway are needed).

Response:

We agree with the reviewer that the sentence is possibly overstated. We have now reworded it as suggested.

Concern #5: The results represent well designed flow cytometry study, however, the measured parameters are limited. In the future, it would have been further interesting if the effect of examined compounds, most of all, the new 4,5-dihydroxy-3,3˘,4˘-trimethoxybibenzyl derivative, will be checked on other parameters, like the production of molecules responsible for killing of pathogens (H2O2, NO) or the phagocytic ability of human monocytes.

Response:

We thank the reviewer for the suggestion. As it has been shown in the results section, different monocyte subsets have different potential of producing inflammatory cytokine/mediators, and displayed different responses to the tested compounds. It is known that different monocyte subsets possess unique phenotypes and functions (Cros et al. Immunity 2010). Thus, as also suggested by the reviewer, it needs to be precisely investigated how each monocyte subset responses to the treatment with the target compound. We plan to apply high dimensional cell profiling technology such as mass cytometry (assessment of up to 40 markers on one cell: Böttcher et al. Nat. Neurosci. 2019, and Sankowski*, Böttcher* et al. Nat. Neurosci. 2019), in order to achieve comprehensive information on drug-related phenotypic and functional changes, as well as cell signaling. 

Concern #6: The authors order is different in “Orders of Authors” section and in the manuscript text.

Response:

We apologize for the mistake. The author order should be as addressed in the manuscript.

Reviewer #2: 

The fact is that macrophages, an important population of cells, are involved primarily in defense against infectious disease and play a central role in maintaining all tissues in a healthy state. On the other hand, macrophages also have a multifaceted role in diseases such as cancer, Alzheimer disease, multiple sclerosis and type 1 diabetes… I think that the topic is interesting, the introduction is sufficient to understand the issue, but that the results within the text should be confirmed by statistics. The methods are well described as well as the presentation of the obtained results (Figures). However, I would like to see more data confirming the macrophage function after treatment with isolated components from a Thai orchid Dendrobium lindleyi Steud such as phagocytosis ability, confirmation of cytotoxicity in co-culture with tumor cells and measurement of NO and H2O2 levels as well as confirmation of anti-inflammatory effect of components through COX-2 inhibition and LOX activities. I believe that confirming the functional ability of macrophages could better link the importance of secretion of individual cytokines with respect to cytokine pleiotropy and redundancy and better and more clearly confirm their immunomodulatory role in health or disease. By confirming the functional capacity of macrophages, the whole research would confirm the greater possibility of application of these components and better justify the mentioned project in plant protection. Such a presentation gives only possible assumptions without confirming the true immunomodulatory role of the isolated components.

Response:

We appreciate the reviewer’s comments and also agree that the conclusion of the study might give an impression of being overstated, as it has been also mentioned by the reviewer #1. We have therefore reworded the main text, where it is required, in order to be less speculative. Furthermore, we have revised the main text to make it more clear regarding the aim and future perspectives of the project.

As we have addressed in the introduction and the results, in this study we aimed to evaluate immune modulatory effects of the five compounds on circulating monocytes ex vivo. These cells possess different phenotypes and functions, compared to tissue-resident (or tumor-associated/infiltrating) macrophages. Most of available cell culture models provide systems for study monocyte-derived cells, which resemble tissue macrophage phenotypes and functions rather than those of circulating monocytes. Therefore, we first validated the use of low-attachment culture system to mimic circulating nature of monocytes, and thus avoid them to become adherent macrophage. Using this low-attachment culture system, we are able to detect all three subsets of circulating monocytes, which are CD14+CD16-, CD14+CD16+ and CD16+CD14- populations, thus are able to model all monocyte subsets ex vivo. We then further demonstrated immune modulatory effects of the tested compounds on different monocyte subsets. It remains however to further investigate:

1) which monocyte subsets (CD14+ or CD14-) are the ones infiltrating to the tissue (or tumor), and further differentiate to tissue-macrophage (or tumor-associated macrophage)

2) whether monocyte-derived macrophages serve as a direct effector at the site of cancer/diseases, and thus potentially provide therapeutic benefit. Or the compound-related therapeutic effects would rather be provided from circulating monocytes directly in the peripheral blood circulation.

3) whether the observed immune modulatory effects could be maintained after monocyte differentiation to tissue (or tumor-associated) macrophage

Furthermore, the immune modulatory effects of the compounds may also provide beneficial effects in a broader spectrum of inflammation-driven diseases such as Chron’s disease (CD), besides lung cancer. In our previous studies (Ziegler*, Böttcher* et al. Nat. Commun. 2019; Böttcher et al. Sci. Rep. 2019), we have demonstrated phenotypic and functional changes of circulating myeloid cells including monocytes in CD, suggesting their important roles in this pathology. Currently, it is our focus to further investigate (using our low-attachment culture system) whether the compounds (especially the new compound, #1) would also provide therapeutic benefits on circulating monocytes obtained from the peripheral blood of patients with CD (ongoing project). 

In addition, as shown in Fig. 2, different monocyte subsets have different potential of producing inflammatory cytokine/mediators. And it is known that different monocyte subsets possess different migratory potential to the tissue (or site of tumor). Thus, it needs to be further investigated, after exposing to the compounds, which monocyte population would provide more therapeutic benefits (immune modulatory effects vs activation-associated (on-site function)) function such as phagocytosis, cytotoxicity. To prove this assumption, we plan to analyse each monocyte subset after a treatment with the desired compound using single-cell sorting and analysis technology (as have been published by us recently, Böttcher et al. Nat. Neurosci. 2019; Sankowski*, Böttcher* et al. Nat. Neurosci. 2019), as well as using microfluidic cell culture system to tract each cell population. Together, we would need to gain more information regarding potential cell signaling pathway, which may be different from those results from cell line studies, in order to target the right pathway.

Finally, as we have written in discussion: “….Our findings suggest an importance and/or feasibility of using ex vivo studied models of different primary human (immune) cells to estimate the therapeutic potential and/or any potential adverse effects of purified natural products, which will further facilitate the adjustment of developmental potential of each isolated compound in diseases….”, we are confident that this study will encourage the colleagues, whose researches focus on ex vivo (or in vitro) therapeutic screening of natural products, to apply such a model on primary human cells.

Second, minor problems, there are some mistakes about editorial handling, e.g. % must be written close the number. Please harmonize the way of writing degree C. SI units should be used… (e.g. μg/ml should be μg/mL). Please separate the SI unit from the number. In vitro and in vivo should be written in italic...... . Statistical value P should be written in italic and with a capital letter. Abbreviations must be explained the first time they are used, both in the Abstract and again in the main text.

Response:

We thank the reviewer for thoroughly reading our manuscript. In the revised manuscript, we have improved the main text as suggested.

---

## [Decision Letter · Decision Letter 1]

19 Aug 2020

Immune modulatory effect of a novel 4,5-dihydroxy-3,3´,4´-trimethoxybibenzyl from Dendrobium lindleyi

PONE-D-20-02679R1

Dear Dr. Böttcher,

We’re pleased to inform you that your manuscript has been judged scientifically suitable for publication and will be formally accepted for publication once it meets all outstanding technical requirements.

Kind regards,

Horacio Bach

Academic Editor

PLOS ONE

Additional Editor Comments (optional):

Reviewers' comments:

Reviewer's Responses to Questions

**Comments to the Author**

1. If the authors have adequately addressed your comments raised in a previous round of review and you feel that this manuscript is now acceptable for publication, you may indicate that here to bypass the “Comments to the Author” section, enter your conflict of interest statement in the “Confidential to Editor” section, and submit your "Accept" recommendation.

Reviewer #1: All comments have been addressed

Reviewer #2: All comments have been addressed

2. Is the manuscript technically sound, and do the data support the conclusions?

Reviewer #1: Yes

Reviewer #2: Yes

3. Has the statistical analysis been performed appropriately and rigorously? 

Reviewer #1: Yes

Reviewer #2: Yes

4. Have the authors made all data underlying the findings in their manuscript fully available?

Reviewer #1: Yes

Reviewer #2: Yes

5. Is the manuscript presented in an intelligible fashion and written in standard English?

Reviewer #1: Yes

Reviewer #2: Yes

6. Review Comments to the Author

Reviewer #1: Pichayatri Khoonrit et al. presented a revised version of the manuscript entitled “Immune modulatory effect of a novel 4,5-dihydroxy-3,3´,4´-trimethoxybibenzyl from Dendrobium lindleyi” where the potential immunomodulatory properties of compounds isolated from Thai orchid Dendrobium lindleyi were described. The overall quality of the revised version was improved and the comments mentioned by the reviewer #1 were addressed point-by-point in the response letter. The answers provided in the response letter were satisfactory and I suggest the manuscript to be published in PLOS ONE.

Reviewer #2: I thank the authors for the answer, explanation and corrections of the manuscript and I hope that the authors with this model of primary human culture of monocyte cells and with different monocyte subtypes will succeed in achieving the appropriate therapeutic goal. I look forward to their next step in assessing the therapeutic effect of individual monocyte populations.

7. PLOS authors have the option to publish the peer review history of their article (what does this mean?). If published, this will include your full peer review and any attached files.

Reviewer #1: No

Reviewer #2: No

---

## [Editor Report · Acceptance letter]

21 Aug 2020

PONE-D-20-02679R1 

Immune modulatory effect of a novel 4,5-dihydroxy-3,3´,4´-trimethoxybibenzyl from Dendrobium lindleyi 

Dear Dr. Böttcher:

I'm pleased to inform you that your manuscript has been deemed suitable for publication in PLOS ONE. Congratulations! Your manuscript is now with our production department. 

Kind regards, 

on behalf of

Dr. Horacio Bach 

Academic Editor

PLOS ONE